# Redox-Mediated Rewiring of Signalling Pathways: The Role of a Cellular Clock in Brain Health and Disease

**DOI:** 10.3390/antiox12101873

**Published:** 2023-10-17

**Authors:** Filip Vujovic, Claire E. Shepherd, Paul K. Witting, Neil Hunter, Ramin M. Farahani

**Affiliations:** 1IDR/Westmead Institute for Medical Research, Sydney, NSW 2145, Australia; filip.vujovic@sydney.edu.au (F.V.); neil.hunter@health.nsw.gov.au (N.H.); 2School of Medical Sciences, Faculty of Medicine and Health, The University of Sydney, Sydney, NSW 2006, Australia; 3Neuroscience Research Australia, Randwick, NSW 2031, Australia; c.shepherd@neura.edu.au; 4Redox Biology Group, Charles Perkins Centre, Faculty of Medicine and Health, School of Medical Sciences, The University of Sydney, Sydney, NSW 2006, Australia; paul.witting@sydney.edu.au

**Keywords:** brain development, neurodegenerative disorders, redox, mitochondria, cellular clock

## Abstract

Metazoan signalling pathways can be rewired to dampen or amplify the rate of events, such as those that occur in development and aging. Given that a linear network topology restricts the capacity to rewire signalling pathways, such scalability of the pace of biological events suggests the existence of programmable non-linear elements in the underlying signalling pathways. Here, we review the network topology of key signalling pathways with a focus on redox-sensitive proteins, including PTEN and Ras GTPase, that reshape the connectivity profile of signalling pathways in response to an altered redox state. While this network-level impact of redox is achieved by the modulation of individual redox-sensitive proteins, it is the population by these proteins of critical nodes in a network topology of signal transduction pathways that amplifies the impact of redox-mediated reprogramming. We propose that redox-mediated rewiring is essential to regulate the rate of transmission of biological signals, giving rise to a programmable cellular clock that orchestrates the pace of biological phenomena such as development and aging. We further review the evidence that an aberrant redox-mediated modulation of output of the cellular clock contributes to the emergence of pathological conditions affecting the human brain.

## 1. Introduction

Metazoans benefit from a capacity to dampen or amplify the rate of cellular events by reprogramming the underlying signal transduction pathways. Neoteny provides an example of such programmability in the context of development. The term neoteny refers to “the preservation of juvenile characteristics in adulthood”, which can be interpreted as the prolongation of youth [1]. A typical example of neoteny is accelerated sexual maturity of the axolotl combined with a retention of juvenile features, e.g., gills, that enables axolotl to occupy deeper aquatic habitats [2]. However, environmental stressors (e.g., absence of deep water) reprogram developmental cascades to transform axolotl into a terrestrial adult form without any gills and with larger feet which allows the adult form to occupy terrestrial habitats [3,4,5]. Another example of neoteny is prolonged ontogeny of the human brain within the primate lineage leading to an extended retention of juvenile features (e.g., synaptic plasticity) in adult human brain [6,7]. Due to such prolonged development, interactions with external stimuli more effectively reshape the postnatal wiring of the human brain as opposed to the predominance of intrinsic self-organisation of the brain in other primates [8]. Given that metazoan developmental landscape is shaped by decentralised self-organising interactions between individual cells [9,10,11,12,13], the acceleration or deceleration of developmental dynamics suggests the existence of a hypothetical “cellular clock” which determines the rate of cellular events.

The proposed concept of a cellular clock can be developed at multiple levels. At a basic intuitive level, the cellular clock describes the rate of completion of a particular cellular task via the directional flow of information in interconnected signal transduction pathways of an individual cell. In this definition, elements that oppose the directional flow of information (i.e., antagonistic elements) slow down the cellular clock and those that facilitate the directional flow of information accelerate the clock. A more precise definition of the proposed cellular clock requires the application of probability theory. From this perspective, discrete events in relaying an upstream signal are modelled as binary events with a Boolean-valued outcome: success (generation of a downstream signal with a probability of ***p***) or failure (no downstream signal with a probability of ***q*** = 1 − ***p***). The probability of success (***p***) or failure (***q***) in each discrete event is determined by the integration of inputs from the antagonistic elements that oppose the directional flow of information and the activity of the agonistic elements that facilitate the directional flow of information. In this sequence, the probability of producing ***k*** successes (i.e., a downstream signal) in ***n*** independent repetitions of a signalling event determines the length of time required to exceed a lower threshold for the activation of a sequential signalling event. In this model a binomial distribution describes outcomes of ***n*** discrete signalling events and hence the probability of observing ***k*** > [threshold] successes in ***n*** discrete signalling events. By this line of reasoning, the operational principles of the proposed cellular clock can be summarised as follows:

**I.** Completion of a cellular event requires the transmission of an upstream signal via a series of interconnected discrete signalling loops (***S*_1_** to ***S_n_***).

**II.** Each signalling loop (***S_i_***) must be repeated ***n*** times to produce ***k*** successes wherein ***k*** corresponds to a minimum signal intensity (i.e., a threshold) that activates the next (***S_i+_*_1_**) signalling loop. 

**III.** The probability of success (***p***) in a discrete signalling event (***n_i_***) within a signalling loop (***S_i_***) is determined by the integration of synchronised inputs from agonistic and antagonistic elements within the same signalling loop (***S_i_***). 

**IV.** Amplification of the agonistic elements (increased ***p***) or inhibition of the antagonistic elements (decreased ***q***) increases the probability of achieving ***k*** successes in recursive trials of a signalling event (***S_i_***) and therefore accelerates the cellular clock. 

Based on this definition, it becomes apparent that the synchronised convergent activity of agonistic and antagonistic elements is required to alter the probability of success (***p***) or failure (***q***) in each discrete signalling event. Prior to providing a network-level logic for such synchronised generation of agonistic and antagonistic elements, revisiting the concept of neoteny at a molecular level affords key insights into the biological identity of such elements. 

In a molecular context, thyroid hormones and reactive oxygen species (ROS) appear to be the key regulators of reprogrammed development (i.e., neoteny) [14,15,16]. Aside from the genomic effects, thyroid hormones amplify the mitochondrial production of ROS via activation of fatty acid β-oxidation [17]. To this end, the mitochondrial generation of ROS appears to accelerate a “cellular clock” of differentiation [18]. Here, cellular clock describes the rate of progression of molecular cascades that orchestrate differentiation dynamics from initiation to completion. Further corroborating evidence regarding the notion of the redox-mediated reprogramming of differentiation is observed in the development of the brain. Shortly before the closure of the neural tube, neuroepithelial cells cannibalise heme-rich erythroblasts leading to the activation of mitochondrial oxidative phosphorylation [19]. Mitochondrial activation and the resultant generation of free radicals triggers the accelerated differentiation of cannibalistic cells into neurons. Notably, the impact of ROS on the resetting of the proposed cellular clock is not confined to the impact on differentiation. In cycling cells, mitochondrial ROS regulates CDK2 activity by the oxidation of a conserved cysteine residue of the protein [20]. The redox-dependent activation of CDK2 calibrates the rate of progression of the cell cycle to mitochondrial activity. Collectively, the evidence suggests that the redox-mediated reprogramming of signalling cascades resets the proposed cellular clock. Considering the proposed model, the ROS-mediated reprogramming of the cellular clock suggests that some antagonistic elements in signalling cascades are potentially inactivated in a redox-dependent manner. In this presentation we initially provide evidence for cryptic modules within signalling networks that can be accessed and reprogrammed in a redox-dependent manner to vary the pace of cellular events orchestrated via these signalling pathways. 

## 2. A Generic Blueprint for the “Cellular Clock”: The Role of Paradoxical Network Motifs in Programmability of Signalling Pathways

The programmability of the cellular clock is a reference to the scalability of the underlying molecular events that determine the duration of time required to complete a specific cellular task such as cell cycle or differentiation. In cycling cells, for example, there is a reserved capacity to lengthen or shorten the G1 phase of the cell cycle [21]. Likewise, the differentiation landscape can be reprogrammed to accelerate the neuronal differentiation of neural progenitor cells during brain development [19]. These observations suggest the existence of a reserved capacity in the network topology of signalling pathways that drive cellular events. Paradoxical network motifs provide a mechanism to install a reserved capacity in signalling pathways. In the context of biological signalling, “paradoxical components” refer to components of a signalling cascade which exert simultaneous antagonistic effects on downstream mediators [22,23]. While, for an extensive coverage of the topic, readers are referred to a review by Hart and Alon [23], an overview of a paradoxical network motif termed the incoherent feedforward loop (I-FFL) is provided here to aid in developing the role of redox in modulation of the proposed “cellular clock” (Figure 1). In I-FFLs, an upstream stimulus generates two competing signals with opposing effects on a downstream target. In an I1-FFL configuration, for example, an upstream stimulus directly activates a downstream mediator while inhibiting it indirectly via a separate downstream mediator (Figure 1). In this configuration, I1-FFL behaves as a pulse generator wherein the duration and intensity of the generated signal is determined by a delay between the direct stimulation and indirect inhibition of the downstream signal [24]. Apart from the programmability of individual I-FFL signals, a series of connected I1-FFLs (I1-FFL_1_, …, I1-FFl_n_), that elicit successive pulses of biological signals in a precise temporal and hierarchical order, can be employed to drive complex biological phenomena such as sporulation [25].

The pace of actions communicated by intracellular signalling pathways linked via multiple programmable I1-FFLs (i.e., cellular clock) will be determined by the pulsed activities of individual I1-FFLs transmitting the corresponding signal (Figure 1). In an individual I1-FFL, while repressing the indirect inhibitory arm will amplify the duration and amplitude of an I1-FFL pulse, activation of the inhibitory arm will have the opposite effect of dampening the pulsed signal. In the discussion that follows critical I1-FFL loops that populate various signalling pathways are highlighted. Further, evidence is provided that key components of I1-FFL loops in metazoan signalling cascades are regulated by redox-dependent mechanisms, foreshadowing the role of redox as a master-regulator of the proposed “cellular clock”.

## 3. Metazoan Signal Transduction Pathways

### Small GTPases

Small GTPases are proteins with an intrinsic capacity to hydrolyse guanosine triphosphate (GTP) to guanosine diphosphate (GDP). Accordingly, small GTPases cycle between an active GTP-bound state and an inactive GDP-bound state. The population of these two states by small GTPases is regulated by a combination of the intrinsically slow rate of GDP dissociation and GTP hydrolysis in addition to extrinsic factors that either promote GTP hydrolysis (GTPase-activating proteins: GAPs) or that accelerate the exchange of GDP for GTP (guanine nucleotide exchange factors: GEFs). Aside from the enzymatic regulation of GTPase activation, reactive oxygen species induce the activation of small GTPases by facilitating the dissociation of bound GDP via the formation of an unstable oxygenated GDP adduct [26]. Both Ras and Rap1A are known to be activated by this redox-mediated mechanism [26]. The redox-mediated activation of Ras^GDP^ is initiated by the oxidation of Cys118-SH to Cys118-S triggering the withdrawal of an electron from GDP and producing G^+^-DP. Subsequently, G^+^-DP is converted to G-DP by elimination of H^+^. Formation of G-DP disrupts specific hydrogen bond interactions between the Ras GTPase and its ligand nucleotide [27]. Finally, this destabilised oxidised GDP reacts with ROS to form 5-oxo-GDP, an event that triggers its release from the catalytic site of the GTPase [26]. Further mechanistic details of this phenomenon have been described by Heo et al. elsewhere [27]. After the release of bound GDP, oxidised Ras does not re-associate with free guanine nucleotides and remains in a poised unoccupied state. The reactivation of oxidised Ras requires free radical scavengers (such as ascorbate and GSH) [26]. Aside from Ras and Rap1A, the small GTPase Rac1 is also regulated by a similar mechanism in a redox-dependent manner [26]. Hence, the redox-mediated regulation of small GTPases occurs in a biphasic manner via an initial deactivation step which primes the enzyme for subsequent reactivation. The importance of this biphasic regulation mode becomes apparent by focusing on network-level interactions of small GTPases. 

Lipid-anchored Ras is activated by Son of Sevenless (SOS) [28] which operates downstream to receptor tyrosine kinases [29]. Interestingly, Ras^GTP^ functions both as a downstream mediator and an allosteric activator of SOS [30]. To provide positive feedback to SOS, Ras^GTP^ binds to the interface between the REM domain and the cdc25 domain of SOS. The resultant increase in interfacial interactions at the active site of the kinase as well as the decreased flexibility of the SOS molecule underpin Ras^GTP^-mediated increases in the catalytic efficiency of SOS. This generates a positive feedback cycle which triggers a sharp amplification of Ras signalling, transforming the analogue signalling output of Ras to a digital binary output [31] (Figure 2). After association with GTP, Ras activates the Raf/MEK/ERK kinase signalling cascade [32], which contributes to the establishment of a pro-anabolic state by mechanisms such as the phosphorylation-mediated stabilisation of c-Myc [33]. The Ras^GTP^-mediated activation of Raf is a complex process that proceeds via the membrane recruitment of Raf, displacement of 14-3-3 protein from the CR2 site of Raf and subsequent dimerisation and phosphorylation of the Raf kinase domain [34]. Ras^GTP^ also binds to the p110 catalytic subunit of class I PI3K which amplifies the production of PIP3 by this enzyme [35,36] (Figure 2). This leads to the emergence of a coherent type-I feed-forward loop (C1-FFL), whereby PI3K receives a dual signalling input from receptor tyrosine kinase (RTK) and from RTK-activated Ras^GTP^. The activation of PI3k/Akt via this dual signalling input complements the pro-anabolic activity of Ras^GTP^/c-Myc axis by the phosphorylation-mediated destabilisation of TSC-2 [37] and the resultant mTOR-dependent inhibition of autophagy. Therefore, the redox-mediated biphasic activation of small GTPases in an oxidising milieu would complement the activation of these downstream mediators via RTKs in response to extracellular signals such as growth factors. However, such complementation will only occur after transitioning from a poised unoccupied state to a GTP-bound state in the presence of sufficient quantities of reducing agents. While the detailed mechanistic aspects relating to the regulation of the thiol status of proteins via glutathione have been reviewed elsewhere [38,39,40], an overview is presented herein to shed light on the critical role of the biphasic regulation of small GTPases discussed above.

Thiol-reduced glutathione (GSH) is consumed in the process of scavenging free radicals and maintaining the thiol status of proteins. The re-establishment of a reduced GSH pool can occur by the de novo synthesis of the moiety or by NADPH-dependent reduction of the oxidised glutathione. The rate-limiting step in GSH biosynthesis is catalysed by glutamate cysteine ligase (GCL) which is composed of a catalytic (GCLC) and modifier (GCLM) subunit. GCLM lowers the K_m_ of GCL for glutamate and raises the K_i_ for the negative inhibitory feedback provided by GSH [41]. Insulin appears to activate GCLC via both IRS/PI3K [42] and Raf/MEK/ERK pathways [43]. The activation of GCLC by insulin could potentially balance the activity of this hormone in stimulating the mitochondrial electron transport chain [44] with the resultant generation of oxidising reactive oxygen species [45,46]. Further, the trans-activation of the GCLC locus is regulated via proximal and distal antioxidant response elements that respond to stressors via NF-κB or AP-1 [47]. Finally, c-Myc stimulates the expression of GCLC through binding to noncanonical E-box motifs in the GCLC promoter [48]. Collectively, the evidence suggests that the catalytic activity of GCLC (and hence the level of GSH) is stimulated in a pro-anabolic state (by insulin and c-Myc) or as an adaptive response to oxidative stress (via the NF-κB pathway). It can be argued that not only the induction of a poised oxidised state of small GTPases requires an oxidising milieu, but also the subsequent transition to a reduced GTP-bound state is driven by the redox-mediated upregulation of GSH biosynthesis. However, the trans-activation of GSH in an oxidising milieu occurs in parallel to the depletion of this moiety due to oxidation. Therefore, it is anticipated that the localised restricted production of oxidising cues (e.g., via platelet-derived growth factor [49]) would amplify the signalling activity of small GTPases [50]. However, the depletion of GSH due to cysteine starvation [51] or sustained oxidative stress (e.g., in hypoxia [52]) is expected to generate a pool of GDP-free but inactive small GTPases that remain in a poised state until the re-establishment of a physiological redox state and accumulation of GSH, upon which the reduced form will bind to GTP to amplify the pro-anabolic signalling output. 

The redox-mediated activation of the small GTPase Ras has a further consequence for the crosstalk between class I PI3K and MAPK pathways. It is known that the PI3K downstream mediator, PKB/Akt, inhibits the Raf/MEK/ERK kinase cascade [53]. The integration of Ras and class I PI3K [54] into the downstream effector Raf results in an incoherent type-I feed-forward loop (Figure 2) in which the RTK-mediated activation of Ras (via Grb2/SOS) positively regulates Raf, while the PI3K-mediated activation of PKB/Akt inhibits it. This topology suggests that MAPK will operate in a pulsatile mode [55,56], with the emergence and decay of the MAPK signals triggered by Ras (the agonistic arm of I1-FFL) and PI3K (the inhibitory arm of I1-FFL), respectively. The biological importance of this pulsatile mode of signalling is foreshadowed by observations that the pulse frequency of MAPK directs the resolution of competing cell fates, e.g., proliferation versus differentiation [56,57]. While the delayed activation of PI3K due to configuration of the RTK/PI3K/Ras axis in a coherent type-1 feed-forward loop [24] de-synchronises outputs of the PI3K and Ras pathways [58] to allow for the generation of MAPK signals (Figure 2), this may be insufficient for full pulsatile activation of the MAPK pathway. Accordingly, redox-mediated amplification of Ras signalling and the downstream ERK signalling [59,60,61,62] (Figure 2) may be necessary to complement the impact of desynchronised crosstalk between PI3K and Ras on activating MAPK signalling output. Accordingly, it becomes of interest whether the microanatomical compartmentalisation of H-Ras GTPase to endocytic vesicles, that activates downstream Raf-1 signalling [63], is assisted by NADPH oxidase-dependent ROS production within the endosomal compartment [64] to resolve the I1-FFL^PI3K/Ras^.

It is noteworthy that redox-mediated activation of small GTPases is not restricted to the Ras family of small GTPases. Exposure to cysteine oxidants increases levels of Rheb^GTP^ [65], a small GTPase which functions as an allosteric activator of the mechanistic target of rapamycin complex 1 (mTORC1) [66]. mTOR protein kinases are closely related homologues of PI3K which play a central role in homeostasis and adaptation to stressors [67]. While mTOR complex 1 regulates cellular homeostatic processes such as autophagy and adaptation to stressors, mTOR complex 2 modulates cytoskeletal reorganisation. To perform these tasks, mTORC1 senses amino acid availability, growth factors, glucose availability and oxidative stress, whereas mTORC2 mainly responds to growth factors via the PI3K/Akt pathway [68,69]. Upon activation, phosphorylation of Akt by mTORC2 complements PDK1-mediated phosphorylation of this kinase, leading to its full activation [70].

The key upstream mediator of mTORC1 is the GTPase Rheb^GTP^ which operates by re-aligning the active-site residues within the mTOR complex into an optimal configuration for catalysis, increasing the catalytic activity of the enzyme by 30-fold [66]. The redox-mediated activation of mTORC1 overrides the input from nutrient availability. Accordingly, the treatment of cells with an oxidising agent renders mTORC1 constitutively active, leading to S6K1 phosphorylation even in a nutrient-depleted state [70]. The enrichment of Rheb^GTP^ in an oxidising milieu appears to be the underlying cause for such an altered interpretation by mTORC1 of nutrient availability [66]. The importance of the redox-mediated activation of mTORC1 in a nutrient-depleted state remains largely unknown. Therefore, only an inference can be made regarding the significance of the redox-mediated regulation of mTORC1, as follows: In a normal physiological context, mTOR activity is regulated by the tuberous sclerosis complex (TSC) composed of TSC1, TSC2 and TBC1D7. In this complex, TSC2 functions as a GTPase-activating protein (GAP) for Rheb leading to the formation of an inactive GDP-bound Rheb [71,72,73]. GAP activity of TSC is regulated by phosphorylation; while Akt phosphorylation inhibits TSC2 [37], the GSK3β phosphorylation of TSC amplifies the inhibition of the mTOR complex in an AMPK-dependent manner [74]. Interestingly, phosphatase and tensin homolog (PTEN), which functions as a negative regulator of the PI3K-Akt signalling pathway, is inhibited by reactive oxygen species [75]. Therefore, the transition to an oxidising milieu (enrichment of ROS) not only activates Rheb directly, but it is also expected to amplify mTOR signalling activity by inhibiting PTEN leading to the activation of Akt. The redox-mediated regulation of mTOR complex becomes critical in controlling the adoption of the rivalling fates of stemness and differentiation by progenitor cells. Emerging evidence suggests that the fine tuning of mTOR signalling is essential to prevent the premature differentiation of neural progenitor cells during development [76,77]. To this end, the enrichment of NADH in progenitor cells [78], owing to the metabolic reliance on glycolysis as opposed to oxidative phosphorylation [79], leads to a shift to a reductive environment [80]. Therefore, it can be stipulated that a reductive environment could favour the maintenance of stem cell status by repressing activity of the Rheb/mTOR pathway in a redox-dependent manner [81]. 

## 4. PI3K/Akt Pathway

The PI3K pathway functions downstream to receptor tyrosine kinases (RTKs), as reviewed elsewhere [82]. In summary, the activation of RTKs by growth factors stimulates PI3K to convert phosphatidylinositol (4,5)-bisphosphate (PIP2) to phosphatidylinositol (3,4,5)-trisphosphate (PIP3). PIP3 then recruits PKB/Akt, allowing PDK1 to phosphorylate Thr308 and to partially activate PKB/Akt [83]. Akt phosphorylates multiple downstream substrates including tuberous sclerosis protein 2 (TSC2) leading to the activation of mTOR complex 1 (mTORC1) [37]. Interestingly, mTOR is not just a downstream substrate of Akt. Full activation of Akt requires phosphorylation of Ser473 of the protein by mTOR [70] or by DNA-PK [84]. The activity of PI3K/Akt is regulated by phosphatase and tensin homolog (PTEN). PTEN is a phosphatase that catalyses the removal of the 3’ phosphate of PIP3 to generate PIP2 upon recruitment to the plasma membrane [85]. While readers are referred to recent reviews on PTEN [86,87], a summary of its regulation is provided here. PTEN is composed of an N-terminal phosphatase domain and a C2 domain which orchestrate membrane association by electrostatic interactions, and a C-terminal tail region whose phosphorylation shields the cationic residues reducing the membrane affinity of PTEN [85]. Shielding of the C-terminal region by phosphorylation via GSK3β and CK2 [88] serves an additional function, that is to prevent the proteasomal degradation of PTEN [89], thus reserving a cytoplasmic inactive pool of the protein poised for recruitment to the cell membrane [85,90]. This reserved pool of PTEN can be accessed through the protein phosphatase 2A-mediated dephosphorylation of Ser380, Thr382 and Thr383 residues within the C-terminal tail of PTEN [91], leading to membrane recruitment and activation of the phosphatase activity. Aside from dephosphorylation via protein phosphatase 2A, PTEN appears to be activated by auto-dephosphorylation [92].

These interactions inform an elemental blueprint for the redox-mediated regulation of the PI3K/Akt signalling pathway by modulating PTEN activity (Figure 3). While PTEN can be regulated at a transcriptional level [93], post-translational mechanisms seem to be the dominant mode of regulation in the rapid adaptation to acute stimuli such as exposure of a cell to growth factors [85]. To this end, the cis-activation of receptor-associated tyrosine kinases by growth factors and other stimuli triggers the recruitment of PI3K and a parallel activation of protein phosphatase 2A [94,95] (Figure 3). PI3K catalyses the conversion of PIP2 to PIP3 which prompts Akt signalling. The catalytic activity of protein phosphatase 2A, on the other hand, activates PTEN [91] which antagonises the function of PI3K by converting PIP3 to PIP2. The depletion of PIP3 by this mechanism prevents the membrane recruitment and activation of PDK1 and PKB/Akt [82]. Aside from this indirect interaction, the upstream activator of PTEN, protein phosphatase 2A, contributes to the inhibition of Akt by the dephosphorylation of Thr-308 of the protein [96] (Figure 2). Therefore, it is essential to uncouple the RTK-mediated activation of PTEN and PI3K to invoke downstream signalling by Akt. In an oxidising milieu, PTEN becomes reversibly inactivated due to the formation of an intramolecular disulfide between the essential active Cys-124 residue and Cys-71 [97]. In vitro, PTEN is inactivated within 10 min of exposure to H_2_O_2_, followed by conversion to the reduced active form over the next 120 min [98]. The redox-mediated inactivation of PTEN leads to elevation of PIP3 and the downstream mediator Akt [98]. It is likely that redox-mediated inactivation of PTEN is essential to uncouple the RTK-mediated activation of PTEN from the kinase activity of PI3K and to enable efficient downstream signalling by Akt. To this end, Oatey et al. demonstrated that while insulin and PDGF both cause the equivalent recruitment of PI3K to the cell membrane, only insulin causes the activation of downstream targets of PI3K products such as Akt [99]. It is curious whether the insulin-mediated redistribution of Glut-4 to the plasma membrane [100] and the resultant activation of the mitochondrial electron transport chain and the production of ROS [101] inactivates PTEN, leading to a resolution of the I1-FFL and activation of Akt.

## 5. Wnt/β-Catenin Pathway

β-catenin is a cytoskeletal protein that stabilises the association of adherens junctions and the cytoskeleton. As a free cytoplasmic moiety, β-catenin has a short half-life of ≈1 h [102] after which it encounters two rivalling fates: proteasomal degradation after phosphorylation by GSK3β [103] or shuttling to the nucleus where it associates with TCF3/LEF1 to function as a transcription factor [104]. To degrade free β-catenin, a destruction complex is assembled consisting of Axin, APC, GSK3β and CK1α which primes the protein for recognition and destruction by the proteasome [105]. The activity of the latter destruction complex is abolished upon the binding of Wnt to its receptors Frizzled and LRP (i.e., the Wnt^on^ state). In a Wnt^on^ state, the recruitment of the scaffolding protein, Dishevelled (Dvl), disrupts the GSK3β-mediated phosphorylation of β-catenin leading to an accumulation of the stabilised cytoplasmic protein. Notably, the activity of Dvl is redox dependent. This is because Dvl binds to the thioredoxin-like protein Nucleoredoxin (NRX), an interaction that reduces the availability of Dvl in a Wnt^on^ state [106]. In an oxidising milieu, Dvl is released from NRX which amplifies the output of the Wnt/β-catenin signalling pathway [106]. The transcription factor c-Myc is one of the key genes trans-activated by β-catenin signalling [107]. c-Myc is a master regulator of pro-anabolic flux. To this end, c-Myc binds to E-box sequences in the promoters of active rDNA clusters and regulates the RNAPI-mediated transcription of 18S, 5.8S and 28S rRNAs [108]. The transcription factor also stimulates the RNAPII-mediated transcription of ribosomal proteins [109]. Further, translation initiation factors, namely, eIF4E, eIF2α, eIF4AI and eIF4GI, are regulated by transcriptional activity of c-Myc [110]. Notably, c-Myc increases the level GSH by trans-activating GCLC [48], enabling the reduction and activation of small GTPases as discussed in a corresponding section. The redox-mediated release of Dvl from a reserve NRX-bound pool and the resultant amplification of the Wnt/β-catenin pathway become critical at higher cellular densities. This is because at high cell densities the balance of competition between the adhesion and signalling activities of β-catenin is tipped in favour of sequestering the β-catenin at adhesion complexes [111]. In such a circumstance, the synchronised release of cadherin-bound β-catenin along with the recruitment of the reserve pool of Dvl would efficiently amplify downstream pro-anabolic signalling. In support of this notion, multiple lines of evidence indicate that the activity of Snail (a master regulator of epithelial-mesenchymal transformation) increases in an oxidising milieu [112,113] leading to the down-regulation of cadherins [114]. Readers are referred elsewhere for an in-depth coverage of this topic [115]. It is noteworthy that the redox-mediated amplification of Wnt/β-catenin occurs in a biphasic manner similar in concept to the mechanism described for small GTPases.

## 6. Notch Signalling Pathway

In the Notch signalling pathway, the transmembrane protein interacts with its ligand (members of Delta/Serrate/Lag (DSL) type) presented on a neighbouring cell leading to the release of the Notch intracellular domain (NICD) [116]. NICD is then shuttled into the nucleus where it binds to the CSL family of DNA-binding proteins and activates transcription of targeted genes [117]. The Notch signalling pathway reshapes the metabolic landscape of cycling cells by two broad mechanisms. Nuclear NICD upregulates the translational capacity of cells by trans-activating the c-Myc gene [118]. This triggers a pro-anabolic shift as Myc drives ribosome biogenesis and enhances global protein synthesis [108,119]. In parallel to empowering ribosome biogenesis, NICD enhances mitochondrial oxidative phosphorylation and generation of mitochondrial reactive oxygen species ROS (mtROS) by association with nuclear and mitochondrial genes that encode respiratory chain components [120,121]. In this manner, signals generated downstream to the Notch pathway align the enhanced anabolic activity to mitochondrial energetic output. This pro-anabolic burst is terminated by a mitochondrial heat flux which destabilises the Notch transcriptional complex switching off the signalling pathway in a temperature-dependent manner [122]. 

While Notch1 is not directly regulated by redox-dependent mechanisms, the upstream and downstream mediators of the Notch pathway operate in a redox-dependent manner. The impact of redox on the Notch signalling pathway occurs at both post-translational and post-transcriptional levels. Central to the Notch signalling pathway is the activity of GSK-3β which regulates the half-life of cleaved intracellular Notch1 (Notch1^IC^) by phosphorylating this protein [123]. The inhibition of GSK-3β shortens the half-life of Notch1^IC^, whereas the activation of GSK-3β reduces the proteasomal degradation of Notch1^IC^. Given that Akt inhibits GSK-3β by the phosphorylation of Ser-9 [124], the ROS-mediated inactivation of the Akt inhibitor, PTEN, could potentially amplify the inhibitory cross-talk between Akt and GSK-3β, shortening the half-life of Notch1^IC^. Another enzyme that regulates the stability of Notch1 is NAD^+^-dependent deacetylase Sirt1 [125]. While the acetylation of the Notch1 intracellular domain (NICD) on conserved lysine residues stabilises this protein and increases its half-life, deacetylation by Sirt1 opposes this effect and destabilises Notch1 [125]. Therefore, the conversion of NADH to NAD^+^ in an oxidising environment [80] would enhance the activity of Sirt1 leading to the destabilisation of Notch1 [125]. Further, AMPK that is activated by ROS [126] amplifies the synthesis of NAD^+^ to activate Sirt1 [127]. At a post-transcriptional level, the availability of Notch1 mRNA is regulated by RNA editing and subsequent nonsense-mediated decay [128]. Hence, the inactivation of nonsense-mediated decay in an oxidising milieu [129] is expected to enhance the availability of Notch1 transcript in the anticipation of transition to an ensuing pro-anabolic state. Therefore, the redox-mediated regulation of Notch1 signalling appears to diverge from other signalling cascades in that an oxidising environment activates Notch antagonistic modulators, Sirt1 and GSK-3β. This inference is supported by network-level interactions of the Notch signalling pathway. It is known that Notch-1 and PTEN operate synergistically [130]. The negative regulation of both PTEN and Notch signalling pathways in an oxidising environment would complement the redox-mediated positive regulation of PTEN and Notch-1 antagonistic cascades such as PI3K, Ras/MAPK and the Wnt/β-catenin pathway [131]. 

## 7. Hypoxia-Inducible Factor (HIF)

Hypoxia-inducible factor 1 (HIF-1) is a heterodimeric DNA-binding complex composed of constitutive HIF-1β and one of either hypoxia-inducible α-subunits, HIF-1α or HIF-2α [132]. In this complex, HIF-1α responds to oxygen availability. While in normoxia, HIF-1α is rapidly degraded [133], a reduction of oxygen tension stabilises the protein [134]. HIF signalling activates genes that are required for adaptation to hypoxia. For example, the HIF-mediated expression of pyruvate dehydrogenase kinase results in the inhibition of pyruvate dehydrogenase preventing conversion of pyruvate into acetyl-CoA which leads to repression of the TCA cycle [135]. The main outcome of HIF signalling is the modulation of mitochondrial dynamics and metabolism to minimise the production of ROS under hypoxic conditions.

Hence, it is not surprising that oxidising agents stabilise HIF-1α in normoxia [136,137,138]. Likewise, a reduced capacity to dismutate ROS leads to the stabilisation of HIF-1α and activation of the HIF signalling pathway [139,140]. However, the structural basis for the ROS-mediated activation of HIF-1α remains largely unknown.

## 8. JAK/STAT Pathway

The JAK/STAT pathway is activated upon the binding of cytokines to cognate cell-surface receptors which leads to the recruitment of intracellular Janus kinases (JAKs) and the subsequent trans-phosphorylation. Downstream STATs are then phosphorylated by trans-phosphorylated JAKs (Tyr701 in STAT1 and Tyr705 in STAT3), leading to nuclear shuttling, association with specific enhancers and activation of the target genes [141,142]. Aside from the canonical regulation, STATs are phosphorylated by RTKs such as EGFR and PDGFR, and by non-receptor tyrosine kinases such as Src kinase and ABL [143]. ROS exposure has been shown to activate STAT1 and STAT3 within minutes [144,145]. As for the redox-mediated regulation of HIF, the structural basis for the ROS-mediated activation of STAT1 and STAT3 remains to be elucidated.

## 9. NF-κB Signalling Pathway

The nuclear factor-κB (NF-κB) family of transcription factors comprise five members: NF-κB1, NF-κB2, RelA, RelB and c-Rel, that control diverse facets of immune response and inflammation [146] and development [147]. NF-κB transcription factors reside in an inactive state within cells poised for activation. To maintain the poised state, a family of inhibitory proteins including IκB family members bind to NF-κB proteins and sequester these within the cytoplasm [146]. Upstream signalling leads to the phosphorylation of IκBα by IκB kinase (IKK) complex. The phosphorylation of IκBα triggers the ubiquitination and proteasomal degradation of the protein [148]. In consequence, the free NF-κB complex is shuttled into the nucleus where it triggers transcription of the target genes [149]. Aside from canonical induction, a non-canonical pathway for the activation of NF-κB has been described which initiates upon p100 phosphorylation by NF-κB-inducing kinase (NIK) [150]. The subsequent ubiquitination of p100 and degradation of its C-terminal peptide [151] generates NF-κB2 p52 followed by its shuttling to the nucleus. Apart from enzymatic regulation, ROS can also stimulate the activation of NF-κB in some cell types [152,153]. However, this finding is not robust and is not detectable in all cell types and conditions. Further, crosstalk between ROS and NF-κB pathway mediators appears to be complex, resulting in distinct outcomes in the cytoplasm and nucleus [154]. The inhibition of DNA-binding activity of NF-κB [155] is triggered by oxidation of Cys-62 of p50 [156,157]. ROS also affects upstream mediators of the NF-κB pathway. Exposure to H_2_O_2_, for example, triggers the phosphorylation of specific tyrosine residues of IκBα [158,159,160], leading to the release of NF-κB and its shuttling into the nucleus [161]. The inconsistencies in reports detailing the redox-mediated regulation of NF-κB could potentially be attributed to the context and mode of activation of this signalling pathway. For example, ROS-mediated inhibition of proteasome increases the apparent half-life of ubiquitinated IκBα leading to the reduced nuclear shuttling of NF-κB [162,163]. Another observation which attests to inherent challenges in the interpretation of the crosstalk between ROS and the NF-κB signalling pathway is the redox-mediated regulation of IKKβ [164]. The ROS-mediated S-glutathionation of the Cys179 of IKKβ leads to a reversible inhibition of its kinase activity, an effect that is amplified by deficiency of glutaredoxin-1 [164]. Further, NF-κB regulates the expression of genes encoding major antioxidant proteins such as manganese superoxide dismutase [165], ferritin heavy chain [166], thioredoxin-1 and thioredoxin-2 [165,167]. Considering the rapid nuclear shuttling of NF-κB which upregulates genes involved in modulating the antioxidant capacity of cells, timing could be critical in measuring the impact of ROS on the output of the NF-κB signalling pathway. One potential scenario is that the rapid reversible inactivation of IKKβ and its subsequent activation via reducing agents provides a short temporal window for the redox-mediated rewiring of other signalling cascades prior to the nuclear translocation of NF-κB which upregulates the expression of antioxidant genes. 

## 10. The Ubiquitin System

The ubiquitin-proteasome system (UPS) orchestrates the selective elimination of eukaryotic proteins [168,169,170]. The selection occurs by enzymatic conjugation of the small protein ubiquitin to specific target proteins [171]. This is followed by the degradation of the targeted protein by the 26S proteasome complex with the release of free and reusable ubiquitin [168]. The interface of the redox state and the cellular clock can be considered from two different perspectives, the role of UPS in eliminating oxidised proteins and the redox-mediated regulation of UPS activity. It is estimated that about 70–80% of oxidised proteins are eliminated via the proteasomal pathway [172]. Therefore, the activity of UPS reduces the pool of oxidised inactive proteins that can be recruited into signalling networks. However, key enzymes in the ubiquitin pathway, the E1, E2 and E3 enzymes, contain active site cysteine residues which can be oxidised. Therefore, it is not surprising that upon transitioning into an oxidising state, the activity of the UPS is significantly reduced [173,174]. 

## 11. A Blueprint for Redox-Mediated Resetting of the Cellular Clock

A key adaptation of signalling pathways in an oxidising milieu is triggered by the inactivation of PTEN due to intramolecular disulfide formation between the essential active Cys-124 residue and Cys-71 [97]. This rapid redox-mediated inhibition of PTEN (minutes) [98] will activate the PI3K/Akt cascade by reducing the reverse conversion of PIP3 to PIP2 [175] (Figure 4). This redox-mediated effect is expected to be more pronounced in situations wherein the PI3K/Akt pathway is active. This is because PIP3 represents <0.05% of the total phosphoinositide in cells and it is the enzymatic activity of PI3K that triggers a rapid 100-fold increase in PIP3 in the inner leaflet of the cell membrane [176,177]. Considering the role of PIP3 in the full activation of Akt [178], it can be argued that the redox-mediated inhibition of PTEN may potentially be a prerequisite for the transmission of RTK signals via the PIP3/Akt axis. In accord with this line of reasoning, it has been demonstrated that platelet-derived growth factor (PDGF) transiently increases the intracellular concentration of hydrogen peroxide (H_2_O_2_) and that neutralising this activity via antioxidants blunts the signalling activity of this growth factor [179]. Likewise, epidermal growth factor-induced intracellular H_2_O_2_ formation is required for the inhibition of protein tyrosine phosphatase activity and for EGF-induced protein tyrosine phosphorylation [180]. Subsequent to the redox-mediated reprogramming of I1-FFL^PI3k/PTEN^, the amplified activity of Akt invokes another I1-FFL. Downstream to RTK signalling, while Ras activates the Raf/MEK/ERK kinase signalling cascade [32], PKB/Akt inhibits the Raf/MEK/ERK kinase cascade [53] (Figure 4). This I1-FFL configuration is amplified by the binding of Ras^GTP^ to the catalytic subunit of PI3K which amplifies the production of PIP3 by this enzyme [35,36] (Figure 4). Two mechanisms could regulate the pulsed output of I1-FFL^Akt/Ras^. The redox-mediated activation of Ras^GTP^ will amplify input from the direct stimulatory arm of the I-FFL. Further, the reversal of the redox-mediated inactivation of PTEN could reduce the inhibitory input by reprogramming I1-FFL^PI3k/PTEN^ and terminating the Akt signals. 

A third I1-FFL emerges due to the opposing effects of Ras and Rap1 on Raf1 (I1-FFL^Ras/Rap1^) [181] (Figure 4). The lipid-anchored Ras is activated by Son of Sevenless (SOS) [28] which operates downstream to RTKs [29]. 

In parallel, RTKs enhance the GEF activity of C3G, leading to activation of the downstream Rap1 GTPase [182]. Activated Rap-1 then associates with and traps the Ras downstream effector Raf-1 in an inactive form [183]. The intensity and duration of the Raf-1 signal in I1-FFL^Ras/Rap1^ configuration will be determined by competing activities of Ras and Raf-1. Therefore, it is expected that the differential subcellular localisation of the latter GTPases [184] would alter the activity of the ERK pathway in these subcellular locations. In support of this notion, it has been demonstrated that EGF induces sustained ERK activity near the plasma membrane in sharp contrast to the transient activity observed in the cytoplasm and nucleus [185]. Aside from differential subcellular localisation, redox-mediated regulation could play a role in regulating the I1-FFL^Ras/Rap1^ signalling output. In particular, ROS generation via mitochondria combined with the provision of GTP has the potential to reprogram the I1-FFL^Ras/Rap1^ signal. In transformed cells, Ras not only localises to mitochondria but also suppresses mitochondrial oxidative phosphorylation and enhances the generation of reactive oxygen species by the organelle [186,187]. Hence, it can be postulated that the redox-mediated reprogramming of I1-FFL^PI3k/PTEN^, I1-FFL^Akt/Ras^ and I1-FFL^Ras/Rap1^ could be a major driver of the proposed cellular clock. 

Two key downstream mediators of PI3k and MAPK pathways are the transcription factors c-Myc [188] and cyclic AMP response element (CRE)-binding protein (CREB) [189]. Upon activation, c-Myc amplifies pro-anabolic flux induced by RNAPI [108] and RNAPII-mediated transcription of ribosomal proteins [109]. CREB complements these effects of c-Myc by regulating the expression of metabolic genes [190,191] and mitochondrial electron transport chain components [192,193]. In parallel, the redox-mediated activation of mTOR via Rheb GTPase amplifies output of the Wnt/β-catenin signalling pathway, and a redox-mediated reprogramming of the NF-κB pathway. Both are expected to contribute to the acceleration of the biochemical events that underpin the proposed “cellular clock”. Signals generated downstream to the mTOR pathway increase the translation efficiency of mRNAs [194]. The mTOR-mediated amplification of translation efficiency combined with the inhibition of autophagic flux [195] are essential for the progression of the cellular clock. This dependence partly stems from the energetics of transcription and translation. Protein synthesis requires ~5 ATP per peptide bond or ~2300 ATP per typical protein synthesised [196]. Aside from translation, transcription is also an energetically costly mechanism [197] with an estimated cost of ≈50 ATPs per nucleotide [197]. Hence, in a pro-anabolic state, periods of relative ATP deficiency could potentially activate AMPK leading to inhibition of mTOR signalling pathway and consequently, enhanced autophagy and reduced translation efficiency and a consequential deceleration of the cellular clock. Redox-mediated amplification of the Wnt/β-catenin signalling pathway trans activates c-Myc [107] by regulating the RNAPI-mediated transcription of 18S, 5.8S, and 28S rRNAs [108] and RNAPII-mediated transcription of ribosomal proteins [109]. Further, translation initiation factors, eIF4E, eIF2α, eIF4AI and eIF4GI, are regulated by the transcriptional activity of c-Myc [110]. Hence, it can be argued that the redox-mediated acceleration of the proposed cellular clock leads to the amplification of the anabolic capacity of cells by two mechanisms, I. enhanced capacity of the translational machinery as outline above and II. Increased availability of ATP. The increased availability of ATP is due to the fact that kinases use ATP as the phosphoryl group donor [198], whereas dephosphorylation by PTEN does not regenerate the consumed ATP. Therefore, the amplified activity of PTEN in I1-FFL^PI3k/PTEN^ is expected not only to delay the downstream signalling events but also to directly deplete the ATP required to fuel these events. This reveals the key operational principle of the proposed cellular clock. We anticipate that acceleration of the cellular clock will reduce the time required for completion of a specific cellular task triggered by the upstream signals. It will also enhance the ATP pool and prepare the translational machinery for synthesis of the required proteins. Manifestations of an accelerated cellular clock are expected to be context dependent. In cycling cells, for example, it is expected to shorten the G1 phase of the cell cycle [21]. Likewise, the redox-mediated resetting of the cellular clock accelerates the neuronal differentiation of neural progenitor cells during brain development [19]. 

It is noteworthy that a redox-mediated clock is anticipated to exhibit bistability. This is because redox-sensitive thiols of key proteins populating the circuitry of the cellular clock remain in a poised state after oxidation and subsequent reduction is necessary for the rapid activation of these proteins. This reliance on a balanced access to oxidising and reducing moieties enables the cell to distinguish between a stressor-mediated shift to a sustained oxidising milieu (e.g., in hypoxia [52]) and a physiological state characterised by the balanced presentation of reducing and oxidising cues. Therefore, it is anticipated that the clock will be arrested in the absence of reducing moieties or a predominance of oxidising cues.

A question that arises from these observations is that if one considers the cellular clock as master regulator of the pace of cellular events, what mechanism then regulates the cellular clock? Given the critical role of redox-sensitive GTPases in the proposed circuitry of the cellular clock, it is not surprising that mitochondria can efficiently reprogram the clock by a synchronised supply of GTP and redox-active moieties [19].

## 12. Mitochondria and the Cellular Clock 

Mitochondria produce GTP as a by-product of citric acid cycle and ROS as a consequence of activity of the electron transport chain [45,46]. Detailed study of the mitochondrial electron transport chain revealed that the ubiquinone binding sites in complex I and complex III, glycerol 3-phosphate dehydrogenase, the flavin in complex I, the electron transferring flavoprotein:Q oxidoreductase of fatty acid beta oxidation and pyruvate and 2-oxoglutarate dehydrogenases, all contribute to electron leakage and ROS generation [45]. Notably, there is considerable variability in contribution of these components to the generation of ROS [45]. For example, in cells with a repressed electron transport chain dihydroxyacetone phosphate is converted to glycerol-3-phosphate as a pro-survival metabolic reprogramming to regenerate NAD^+^ [199]. In this state, oxidation of glycerol-3-phosphate by mitochondrial glycerol-3-phosphate dehydrogenase in the mitochondrial intermembrane space leads to a significant leakage of electrons and the generation of reactive oxygen species [200]. The rate of production of ROS by this mechanism is very high and comparable with the maximum rate of ROS generation reported for complex III when inhibited with antimycin A [200]. Hence, mitochondria appear to have a capacity to reprogram the cellular clock by spatially and temporally regulating the production of ROS and GTP. The localised acceleration of neuronal differentiation by mitochondrial oxidative phosphorylation corroborates this notion [19]. Likewise, it is known that a diminished mitochondrial population and a shift to glycolytic metabolism characterise progenitor cells that continue self-renewal, while the activation of oxidative phosphorylation leads to the differentiation of neural progenitor cells [201]. Another interesting line of evidence is provided from mitochondrial dynamics in Hutchinson–Gilford Progeria (HGPS). HGPS is caused by a spontaneous point mutation in the LMNA gene [202]. HGPS is characterised by accelerated aging (5–10 times faster than normal aging) mimicking phenotypic changes observed in elderly people [203]. A hallmark of HGPS is mitochondrial dysfunction leading to impaired respiration, accumulation of ROS and low ATP levels [204,205,206]. A reversal of mitochondrial dysfunction, on the other hand, alleviated senescence in an in vitro model of HGPS [207]. These lines of evidence raise an important question. Is it conceivable that mitochondria are key drivers of the cellular clock? This proposition is intuitively plausible. Regulation of the cellular clock by mitochondria would align the rate of ATP production by these organelles to the rate of ATP consumption during anabolism, preventing a shift to negative ATP economy. 

## 13. Reprogramming of the Cellular Clock: Application in Precision Medicine

Theoretically, reprogramming the proposed cellular clock could be achieved at two levels. Altered redox status of cells is a potential strategy to accelerate or to decelerate the cellular clock. However, this approach is complicated by inherent challenges in controlling the spatial (i.e., subcellular) and the temporal facets of the redox state. The differential subcellular localisation of Ras and Rap1 GTPases [184] provides a clear example of challenges in regulating the clock by altering the redox state of a cell. The post-transcriptional modulation of the mRNAs encoding key components of the cellular clock is an alternative strategy to regulate the pace of biological events. The post-transcriptional inhibition of PTEN mRNA by microRNA-21 is known to reduce the level of the associated protein leading to an enhanced proliferation rate [208]. MicroRNAs hybridise to other seed regions of PTEN to effectively reduce the post-transcriptional level of the mRNA [209]. Accordingly, employing synthetic miRNAs that target complementary seed regions embedded non-randomly [210] in mRNAs encoding components of the cellular clock could be an effective strategy to reset the clock by altering the output of multiple I1-FFLs synchronously. It must be pointed out that these two strategies could be combined to enhance the effectiveness of resetting the cellular clock and the activity of the heme macrocycle clearly illustrates this point. Heme not only activates the mitochondrial electron transport chain and enhances production of free radicals by mitochondria, but it also facilitates processing of the pri-miRNAs to miRNAs [211,212]. Given the improved understanding of microRNA-target recognition principles [213], designing synthetic microRNA customised to the genomic profile of an individual could potentially be a promising therapeutic strategy in precision medicine aiming at the effective resetting of the cellular clock.

## 14. Implication of Redox-Mediated Regulation of Cellular Clock in Brain Development and Disease 

The proposed “cellular clock” is a reference to the interconnectivity of signalling pathways that determines the pace of biochemical events by controlling the outside-in flow of information (i.e., signals). Thus, it is inferred that the clock is set to a maximum in cycling cells and arrested in differentiated cells. Central to the cellular clock are redox-sensitive proteins that populate key nodes of I1-FFL motifs (e.g., PTEN, Ras). Hence, the redox-mediated regulation/dysregulation of the key elements of the cellular clock is expected to contribute to neurodegenerative disorders. In this context, the underlying network topology of the clock wired to amplify the output of specific signalling pathways in a redox-dependent manner bears relevance to the pathogenesis of neurodegenerative disorders. However, before discussing the role of the cellular clock, we first review the evidence for the role of an oxidising environment (i.e., oxidative stress) in the pathogenesis of neurodegenerative disorders. The association of oxidative stress and neurodegeneration is manifest in Alzheimer’s disease (AD) [214,215]. Interestingly, oxidative stress appears to be an early change in AD, being present in a pre-clinical phase known as mild cognitive impairment that is characterised by a relative absence of senile plaques and neurofibrillary tangles [216,217,218]. In a parallel situation, patients with Down syndrome who are predisposed to early onset AD [219] exhibit significantly higher levels of oxidative stress despite a lack of senile plaques and neurofibrillary tangles [220,221]. The increased oxidative stress in AD is associated with another consistent pathological finding, that is reduced energy metabolism in the affected brain [222,223,224]. Such an association puts the spotlight on mitochondria as the inhibition of electron transport chain components brings about the amplified production of ROS by these organelles [225,226]. Corroborating evidence for the role of mitochondria in AD is provided by a documented deficiency of mitochondrial enzymes such as cytochrome oxidase in AD brains [227,228,229,230]. Several intracellular and extracellular changes in AD could explain the coincidence of amplified ROS generation and repression of the electron transport chain in AD. Upon inhibition of the components of the electron transport chain, carriers upstream from the site of inhibition become fully reduced, leading to the amplified production of ROS [231,232]. Additionally, in order to regenerate NAD^+^ in cells with a repressed electron transport chain, dihydroxyacetone phosphate is converted to glycerol-3-phosphate [199] which triggers the enhanced production of ROS by glycerol-3-phosphate dehydrogenase [45]. Finally, microvascular changes in AD brains with the resultant chronic hypoxia [233] could potentially invoke the increased production of ROS by complex III of the electron transport chain [138,234]. Collectively, evidence suggests that a shift to an oxidising intracellular state, mediated via amplified ROS production, is a well-documented feature of AD. Following this line of reasoning, it becomes curious whether network-level signatures of a reprogrammed cellular clock due to amplified ROS production are detectable in AD brains. To answer this question, first we review how the cellular clock is re-purposed in terminally differentiated neurons (Figure 5). 

The PI3K/Akt pathway orchestrates synapse formation [235,236,237] and modulates synaptic plasticity [238,239]. A chemical synapse can be defined as a presynaptic specialisation for rapid synaptic vesicle recycling via exocytosis and endocytosis, a post-synaptic presentation of neurotransmitter receptors, and an array of adhesion complexes stabilising the spatial association of pre- and post-synaptic terminals. In presynaptic vesicle trafficking, exocytosis is regulated by the activity of PI(4,5)P_2_ which facilitates docking prior to Ca^2+^-triggered fusion [240]. The subsequent endocytosis of the clathrin-coated vesicle requires conversion of PI(4,5)P_2_ to PI(3,4)P_2_ [241]. The dynamic turnover of phosphoinositides in endocytosis provides a concentrated zone of these signalling lipids that are recognised by reader domains of Akt and PTEN, leading to a tightly regulated PI3K/Akt signalling cascade [242]. Interestingly, Akt functions as a cargo adaptor protein during clathrin-mediated endocytosis [243], a role that is unrelated to its kinase activity. Further, PI3K/Ras activity appears to regulate clathrin-independent endocytosis [244,245,246]. The recruitment of PI3K/Akt in the endocytic apparatus makes more sense in light of evidence for the role of actin and dynamin in the scission of the endocytic vesicle [247,248]. The activity of PI3K/Akt regulates downstream Rac1 GTPase, leading to the polymerisation of actin and eventual scission of the endocytic vesicle [249,250]. The role of PI3K/Akt in synaptic transmission of signal is not confined to the presynaptic terminal. In the post-synaptic terminal, Akt phosphorylates the type A gamma-aminobutyric acid receptor, leading to an increased representation of the receptor on the plasma membrane surface and enhanced synaptic transmission in neurons [251] (Figure 5).

Satoh et al. recently added an unexpected twist to the role of PI3K/Akt in synaptic transmission by demonstrating that mitochondria chaperone the endocytic vesicle in the presynaptic terminal and promote its maturation by utilising the voltage-dependent anion channel 2 (VDAC2), a mitochondrial outer membrane protein, to bind to the vesicle-associated PI3K [252]. Aside from the provision of ATP and Ca^2+^, recent findings foreshadowed the unexpected role of mitochondrial ROS in regulating the strength of postsynaptic GABA(A) receptors at the inhibitory synapses of cerebellar stellate cells [253]. To enhance synaptic transmission, ROS production via synaptic mitochondria is upregulated by the recruitment of ATPase inhibitory factor 1 (IF1) [254]. These and other studies (reviewed elsewhere [255]) are gradually reshaping the conventional view that the mitochondrial contribution to synaptic transmission is confined to energetic support of the synaptic machinery. The physiological production of ROS by mitochondria is emerging as a driver of key events that underpin the efficiency (i.e., rate) of synaptic transmission. This late paradigm shift in part reflects ambiguities in framing the role of redox in synaptic activity. Evidence presented herein that cryptic bottlenecks in the network topology of PI3K/Akt/Ras signalling pathways can be resolved by the redox-dependent reprogramming of conserved I-FFLs provides a potential explanation for the role of mitochondrial ROS generation in synaptic transmission. This suggests that the proposed cellular clock undergoes a further adaptation in differentiated neurons to generate a subcellular clock that controls the pace of synaptic transmission. By this line of reasoning and given the enhanced level of ROS in AD, is there any evidence for neuronal hyperexcitability in the course of the disease?

Neuronal hyperexcitability is considered a consistent finding in AD. These episodes of neuronal hyperactivity manifest as non-convulsive epileptic discharges [256], particularly in the early stages of AD [257,258,259]. In animal models of AD, hyperactivity of cortical neurons results from decreased GABAergic inhibitory input [260]. Aside from a reduced activity of the inhibitory parvalbumin interneurons [261], attenuation of the GABAergic input has been linked to a reduced membrane representation of GABA receptors in AD [262], known to be modulated via Akt phosphorylation [251]. These changes are aligned to the hyperactivation of PI3K/Akt [263,264] and redox-mediated repression of the antagonistic PTEN [263,265,266] in AD. It has been demonstrated that the S-nitrosylation of redox-sensitive Cys-71 and Cys-124 of PTEN in early Alzheimer’s disease leads to the amplified degradation of the protein via the ubiquitin–proteasome system [265]. To this end, the outcome of S-nitrosylation contrasts sharply with the reversible modification of the redox-sensitive thiols of PTEN that preserves a cytoplasmic pool of the protein for subsequent recruitment, as discussed in a previous section. Interestingly, while wild-type PTEN alters tau phosphorylation, increases the tau–microtubule association and decreases the formation of tau aggregates, the mutant PTEN increases tau aggregation and impairs tau binding to microtubules [267] (Figure 5). On the other hand, Akt also prevents tau ubiquitination and its subsequent degradation [268]. Hence, the unbalanced redox-mediated activity of PI3K/Akt is expected to trigger major pathological findings in AD. In agreement with this line of reasoning, O’Neil refers to AD as a pathology that stems from the sustained over-activation of neuronal PI3-kinase/Akt [269]. The altered activity of PTEN in neurodegenerative disorders could also be triggered by mutations of other partnering proteins that control the activity of PTEN. One such example is the mutation of DJ-1 (alias: Park7) that manifests as autosomal recessive early-onset parkinsonism [270]. This protein functions as a transcriptional regulator of antioxidative genes [271], and also as a negative regulator of PTEN activity [272]. Another downstream mediator of PTEN is Pink1 [273], mutations of which are strongly associated with early-onset parkinsonism [274]. Aside from the downstream effects of a shift towards an oxidising state, one expects to see a parallel activation of signalling pathways that offset such oxidative stress. As mentioned previously, the NF-κB pathway is a major driver of adaptation to oxidative stress [275]. Consistent with amplified ROS production in AD, Chen et al. found elevated levels of the β-site APP cleaving enzyme 1 (BACE1) and NF-κB p65 in the brain of some AD patients [276,277,278]. This finding is of significance given that BACE1 cleaves β-amyloid precursor protein to generate amyloid β protein, a central component of neuritic plaques in AD brains.

Aside from the depletion of differentiated neurons, diminished adult neurogenesis in AD [279] contributes to the pathogenesis of the disease [280]. Would a shift to an oxidising milieu due to mitochondrial dysfunction amplify the differentiation of neural progenitor cells and lead to the reported depletion of these cells in AD? The role of redox in accelerating neuronal differentiation during brain development [19] suggests that amplified ROS production in AD may trigger the depletion of adult neurogenic precursors by a similar mechanism. To this end, the role of R-loops (RNA-DNA hybrids) in regulating differentiation dynamics [281,282] and in the pathogenesis of AD [283] is worthy of attention. Factors that predispose to R-loop formation (e.g., RNA polymerase-II pause [284]), induce DNA damage, or reduce the capacity to repair such damage accelerate neuronal differentiation dynamics [282]. Amongst these are hallmarks of the mitochondrial dysfunction: decline of ATP and NAD+, and elevation of glycerol-3-phosphate [199]. This is because while the reduction of NAD+ curbs DNA damage repair capacity [285], the metabolisation of glycerol-3-phosphate by mitochondrial glycerol-3-phosphate dehydrogenase leads to a significant generation of ROS [200] which induces DNA damage. Once the neuronal differentiation cascade is initiated, the cellular clock would determine the pace of differentiation by mechanisms such as regulating chromatin remodelling [286], activating neuronal late-phase gene expression [287] and transmitting nerve growth factor downstream signals [288]. Further, retinoic acid, a potent inducer of neuronal differentiation [289,290,291,292,293] appears to switch on the differentiation-related genes via activity of the downstream PI3k/Akt pathway [294]. Hence, the ROS-mediated activation of PI3K/Akt (i.e., inhibition of I1-FFL^PI3K/PTEN^) is anticipated to prompt the accelerated differentiation of neural progenitor cells in AD, contributing to the reported depletion of these cells in the course of the disease [279,280]. Aside from differentiation, activation of a cellular senescence program in AD [295,296,297,298,299] may contribute to the depletion of neural progenitor cells. PTEN in collaboration with the mTOR pathway appears to be the key driver of PTEN-loss-induced cellular senescence [300]. Likewise, the expression of Ras GTPase results in a permanent G1 arrest that is phenotypically indistinguishable from cellular senescence [301,302].

Collectively, evidence leads to a provocative hypothesis: is AD a consequence of the redox-mediated reversal of molecular mechanisms that orchestrate human brain neoteny? The proposed reversal of neoteny is analogous to the ROS-induced reprogramming of tadpole neoteny that triggers an accelerated ontogeny [14], albeit with a key difference. While the developmental cascade is triggered via hormones [14], the pathological cascade in AD is driven cell intrinsically by the dysregulated mitochondrial generation of ROS. In this context, chronic inflammation and the resultant enhanced production of ROS [303], such as that which occurs as a consequence of an altered gut–brain axis [304], could act synergistically with mitochondrial ROS to drive key manifestations of AD.

## 15. Conclusions

While altered redox is generally considered as an outcome of cellular dynamics, we review the evidence for an alternative interpretation that redox-sensitive proteins are central to reprogramming the interconnectivity of various signalling pathways. We propose that several paradoxical motifs (I1-FFLs) in the network topology of PI3K/Akt/Ras signalling pathways restrict downstream signalling as a safeguard mechanism with regulation occurring by the redox-mediated rewiring of the I1-FFLs. To this end, mitochondria play a key role in modulating the cellular redox state which resets the proposed cellular clock. We also propose that the existence of paradoxical network motifs introduces a loophole into metazoan signalling pathways which manifest upon a sustained shift into an oxidised cellular state triggering the major manifestation of the neurodegenerative disorders of the human brain.

## Figures and Tables

**Figure 1 antioxidants-12-01873-f001:**
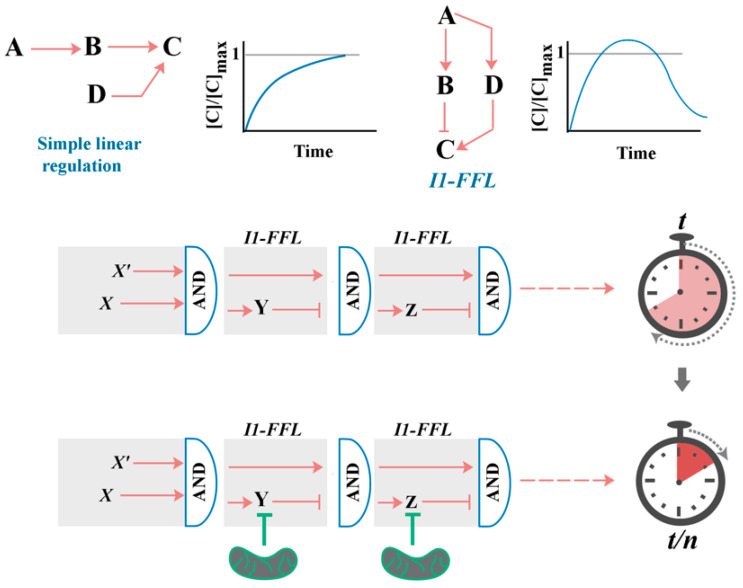
Network motifs utilised in the cellular clock. In an I1-FFL (right), competing inputs from an activator (D) and an inhibitor (B) elicited downstream to a common activator (A) generate a pulsed signalling output (C) as opposed to a linear flow of information (C). Multiple programmable I1-FFLs can be linked via AND gates to control the pace of complex biological phenomena (X, X’, Y, Z represent signalling mediators). Inhibition of the inhibitory arm of I1-FFLs by various cues (e.g., mitochondrial ROS) accelerates the signalling outcome proportional to the number on I1-FFLs linked together in series.

**Figure 2 antioxidants-12-01873-f002:**
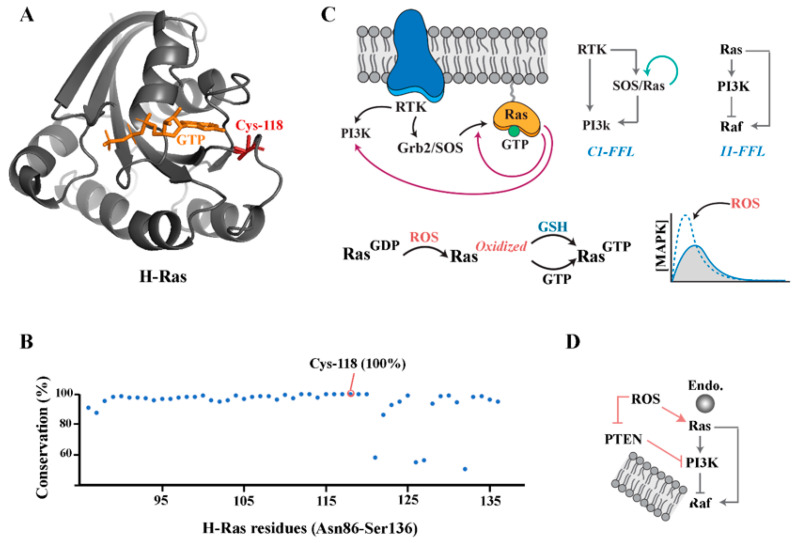
Redox-mediated regulation of I1-FFL^PI3K/Ras^. (**A**) The close-up view of H-Ras (PDB: 121P) shows contribution of the redox-sensitive Cys-118 to the GTP binding pocket of the protein. (**B**) Evolutionary conservation of the redox-sensitive Cys-118 (n = 124 species within the subphylum Vertebrata). (**C**) A simplified presentation of the signals invoked downstream to the receptor tyrosine kinases (RTKs). Note the opposing impacts PI3k and Ras on Raf-1 leading to the emergence of an I1-FFL. In this I1-FFL topology, while RTK-mediated activation of Ras (via Grb2/SOS) positively regulates Raf, RTK/PI3K-mediated activation of PKB/Akt inhibits it. (**D**) Redox-mediated activation of Ras and subcellular localisation of Ras into redox-active endosomes resolves the I1-FFL^PI3K/Ras^.

**Figure 3 antioxidants-12-01873-f003:**
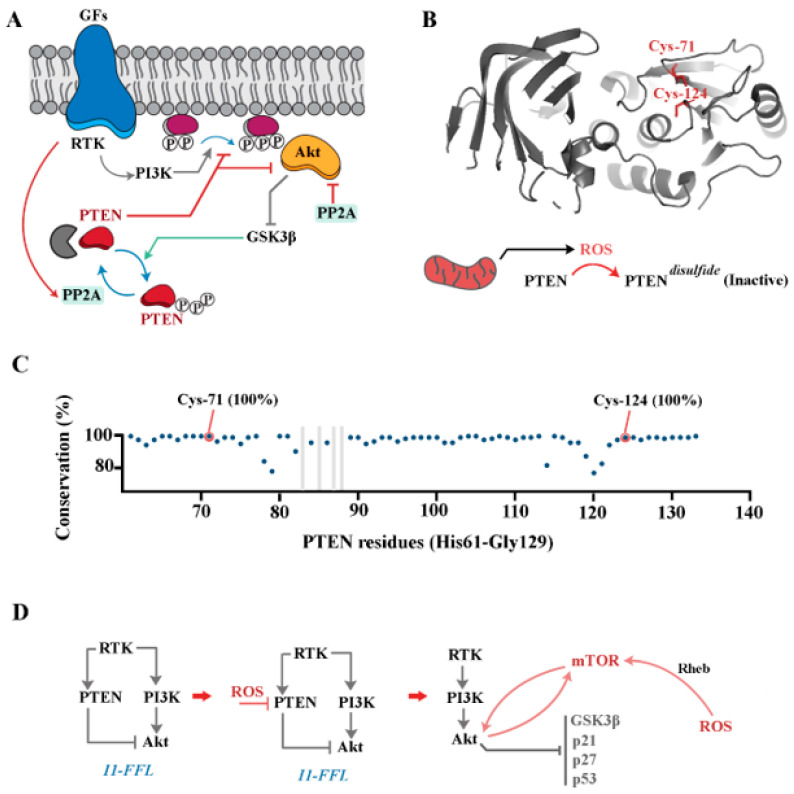
Redox-mediated regulation of I1-FFL^PI3K/PTEN^. (**A**) A simplified presentation of the PI3K and PTEN signals invoked downstream to the activation of RTKs. (**B**) The close-up view of PTEN (AlphaFoldDB: F6KD01) shows the redox-sensitive Cys-71 and Cys-124. (**C**) Evolutionary conservation of the redox-sensitive Cys-71 and Cys-124 of PTEN (n = 124 species within the subphylum vertebrata). (**D**). Redox-mediated resolution of the I1-FFL^PI3K/PTEN^.

**Figure 4 antioxidants-12-01873-f004:**
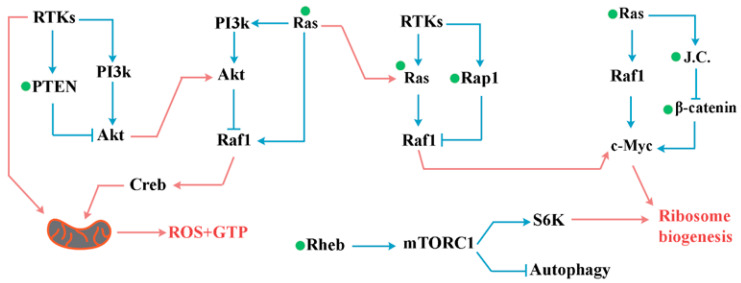
An elemental blueprint for the proposed redox-sensitive cellular clock. The main contributors to a cellular clock are I1-FFL^PI3K/PTEN^, I1-FFL^PI3K/Ras^ and I1-FFL^Ras/Rap1^ connected via AND gates. Reprogramming of key redox-sensitive proteins (marked by green circles) resolves the I1-FFLs leading to amplification of downstream signalling outputs. Other minor contributing I1-FFLs downstream to Ras signalling are also reprogrammed in a redox-dependent manner (right). While Rap-induced Raf-1 stabilises c-Myc, sequestration of β-catenin to Ras-induced formation of the junctional complexes reduces β-catenin-mediated trans-activation of c-Myc locus. Redox-mediated disassembly of the junctional complexes resolves this I1-FFL. The outcome of redox-mediated reprogramming of linked I1-FFLs is activation of the pro-anabolic master regulator c-Myc and mTOR-dependent inhibition of catabolic activity leading to accelerated progression of biological phenomena (e.g., cell cycle).

**Figure 5 antioxidants-12-01873-f005:**
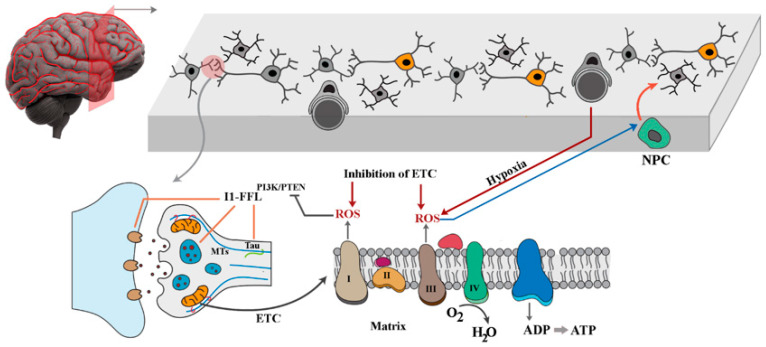
Redox-mediated reprogramming of cellular clock in neurodegenerative disorders. The synaptic transmission of signals, the stability of Tau protein and the associated microtubules and the differentiation propensity of neural progenitor cells (NPC) are controlled by activity of redox-sensitive I1-FFLs (e.g., I1-FFL^PI3K/PTEN^). Hence, ROS generated as a consequence of an impaired electron transport chain (ETC) or hypoxia could potentially amplify the synaptic transmission, reduce tau solubility and trigger accelerated differentiation of neural progenitor cells leading to depletion of these cells.

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
