# Peer review of "Redox-Mediated Rewiring of Signalling Pathways: The Role of a Cellular Clock in Brain Health and Disease"

_antioxidants, 2023, doi:10.3390/antiox12101873_

Round 1

Reviewer 1 Report

Filip Vujovic and co-workers present in their review article in which they have described ROS signaling pathways. The manuscript is interesting and certainly catches the eye of a wide range of readers. The authors have presented some novel ideas that does bring a fresh point of view into the research field. However, the manuscript is composed of different sections that are not connected to each other; i.e., different signaling pathways are separately listed and the reader may get confused how those are connected to each other and if those pathways are the only pathways involved in ROS signaling. Further, it is not clear why those pathways are selected.

Major improvements needed:

1. The first chapter: “On programmability of a cellular clock: aging and neoteny defined as a spectrum” must be removed completely. I do not see any connection between ROS signaling, juvenile characteristics in adulthood, and accelerated sexual maturity. The topic of this chapter is really poorly chosen and gives an impression that the authors have not completely understood the meaning of ROS signaling. Remove the chapter and the text in it.

2. Please explain comprehensively the concept of “cellular clock”.

3. Page 4, lines 117-119: Describe in detail using a figure how “Redox-mediated activation of RasGDP” occurs. This is not explained in the literature well. Explaining this is essentially important, especially because the authors wrote (page 9. Lines 331,332): “It is noteworthy that the redox-mediated amplification of Wnt/β-catenin 331 occurs in a biphasic manner similar in concept to the mechanism described for small 332 GTPases.”

4. Page 4, lines 130-131: Describe in detail “RasGTP functions both as a downstream mediator, and an allosteric activator of SOS” and how RAS activates BRAF (Page 4, line 134), in the same figure in which you describe redox mediated activation of RAS GDP. These are explained only superficially in the text and in the figure 2.

5. Page 6, lines 187-197, in which the authors describe AKT-mediated regulation of ERK1/2 signaling is interesting and should be described better in the text and in the figure 2.

6. In general, the body text should be supported by figures summarizing the signaling pathways. Like this also the incoherent feedforward loop regulation would become clearer to readers. Also, the cellular clock concept, which is one of the main items of the manuscript, should be described better in the text and in a figure, as mentioned above. Currently, it does not become clear to a reader.

7. Page 15, lines 592-603; Please remove the part that explain ROS modifications in cancer to the chapter 6 or to another relevant chapter. Like this the chapter 7 would be more compact.

8. Page 18, line 755: ….provocative hypothesis: While the hypothesis is interesting, it would be a good idea to mention gut-brain axis, which is hypothesized to be involved in the initiation of AD and PD.

Author Response

We are grateful for the encouraging comments and constructive suggestions provided. We have endeavored to answer the questions and accommodate the suggestion as elaborated in the point-by-point rebuttal that follows.

Major improvements needed:

  1. The first chapter: “On programmability of a cellular clock: aging and neoteny defined as a spectrum” must be removed completely. I do not see any connection between ROS signaling, juvenile characteristics in adulthood, and accelerated sexual maturity. The topic of this chapter is really poorly chosen and gives an impression that the authors have not completely understood the meaning of ROS signaling. Remove the chapter and the text in it.

 A: In agreement with the reviewer’s comment, we have thoroughly revised the section on neoteny two achieve two goals. First, we have removed the emphasis on the association of aging and neoteny to reduce the unnecessary clutter in the text. The revised section is called “Introduction” and aims at introducing and developing the concept of cellular clock. Therefore, prior to introducing the network-level logic of a cellular clock, a coarse organism-level explanation is provided to aid the reader in transitioning to the next section.     

  1. Please explain comprehensively the concept of “cellular clock”.

A: Thank you for this suggestion. We have now provided a precise mathematical definition of the proposed cellular clock in the second paragraph of the introduction section in the revised manuscript. To provide a mathematical explanation, we have used a binomial distribution model to explain the binary outcomes of individual signalling events as success or failure. In this model k successes in a series of n similar events are required to surpass the lower threshold for activation of the next signalling event. As such, while agonists of the event increase the probability of achieving k successes, the antagonistic elements delay this outcome. Therefore, the probability of k successes determines the length of a temporal period that is needed to trigger the next signalling event. This definition calibrates the internal clock against an external observer’s clock.

  1. Page 4, lines 117-119: Describe in detail using a figure how “Redox-mediated activation of RasGDP” occurs. This is not explained in the literature well. Explaining this is essentially important, especially because the authors wrote (page 9. Lines 331,332): “It is noteworthy that the redox-mediated amplification of Wnt/β-catenin 331 occurs in a biphasic manner similar in concept to the mechanism described for small 332 GTPases.”

A: In agreement with the reviewer, we have now expanded the section on redox-mediated activation of the Ras GTPase by adding the following section: “Redox-mediated activation of RasGDP is initiated by oxidation of Cys118-SH to Cys118-S· triggering the withdrawal of an electron from GDP and producing G·+-DP. Subsequently, G·+-DP is converted to G·-DP by elimination of H+. Formation of G·-DP disrupts specific hydrogen bond interactions between the Ras GTPase and its ligand nucleotide. Finally, this destabilised oxidized GDP reacts with ROS to form 5-oxo-GDP, an event that triggers its release from the catalytic site of the GTPase26. Further mechanistic details of this phenomenon have been described by Heo et al. elsewhere”. We have also added a new reference 27 which elegantly describes the mechanistic details of this phenomenon. We did not add an additional figure to accommodate a comment by another reviewer who suggested that we reduce the mechanistic clutter in order to aid the reader to grasp the general message easier. However, we are happy to add a supplementary figure if the revised text is deemed insufficient.    

  1. Page 4, lines 130-131: Describe in detail “RasGTP functions both as a downstream mediator, and an allosteric activator of SOS” and how RAS activates BRAF (Page 4, line 134), in the same figure in which you describe redox mediated activation of RAS GDP. These are explained only superficially in the text and in the figure 2.

A: Thank you for this suggestion. We have now added mechanistic explanation for Ras-mediated activation of SOS. The amended section in the revised manuscript maintains that “To provide positive feedback, RasGTP binds to the interface between the REM domain and the cdc25 domain of SOS. The resultant increase in interfacial interactions at the active site as well as the decreased flexibility of the SOS molecule underpin RasGTP-mediated increase in the catalytic efficiency of SOS”.

We have also added a mechanistic explanation and the associated reference (ref #33) regarding RasGTP-mediated activation of the BRAF to revised manuscript. The revised section maintains that “RasGTP-mediated activation of Raf is a complex process that proceed via membrane recruitment of Raf, displacement of 14-3-3 protein from the CR2 site of Raf and its subsequent dimerization and phosphorylation of the Raf kinase domain”.

  1. Page 6, lines 187-197, in which the authors describe AKT-mediated regulation of ERK1/2 signaling is interesting and should be described better in the text and in the figure 2.

A: In agreement with the reviewer, we have revised this section both in the text and the legend to Fig. 2 to improve the clarity of the section.

  1. In general, the body text should be supported by figures summarizing the signaling pathways. Like this also the incoherent feedforward loop regulation would become clearer to readers. Also, the cellular clock concept, which is one of the main items of the manuscript, should be described better in the text and in a figure, as mentioned above. Currently, it does not become clear to a reader.

A: Thank you for this suggestion. We have expanded the Introduction section of the revised manuscript by adding a precise mathematical definition of the proposed cellular clock (please see second paragraph of the introduction section in the revised manuscript). To provide a mathematical explanation, we have used a binomial distribution model to explain the binary outcomes of individual signalling events as success or failure. In this model k successes in a series of n similar events are required to surpass the lower threshold for activation of the next signalling event. As such, while agonists of the event increase the probability of achieving k successes, the antagonistic elements delay this outcome. Therefore, the probability of k successes determines the length of a temporal period that is needed to trigger the next signalling event. This definition calibrates the internal clock against an external observer’s clock. We did not add an additional figures to accommodate a comment by another reviewer who suggested that we must reduce the mechanistic clutter particularly with reference to description of the signaling pathways in order to aid the reader to grasp the general message easier.

  1. Page 15, lines 592-603; Please remove the part that explain ROS modifications in cancer to the chapter 6 or to another relevant chapter. Like this the chapter 7 would be more compact.

A: We agree that the section on cancers becomes a distraction for the reader. However, we could not easily integrate this section into other chapters. To accommodate this suggestion, we have removed this section in the revised document.  

  1. Page 18, line 755: ….provocative hypothesis: While the hypothesis is interesting, it would be a good idea to mention gut-brain axis, which is hypothesized to be involved in the initiation of AD and PD.

A: Thank you for this suggestion. We have mentioned the role of an altered gut-brain axis in the relevant section of the revised manuscript maintaining that “In this context, chronic inflammation and the resultant enhanced production of ROS, such as occurs as a consequence of an altered gut-brain axis, could act synergistically with mitochondrial ROS to drive key manifestations of AD”.

Reviewer 2 Report

In their review manuscript entitled "Redox-mediated rewiring of signalling pathways: The role of a cellular clock in brain health and disease" Filip Vujovic and colleagues elegantly present a novel and interesting angle on redox-mediated rewiring of signaling by focusing on two important signaling proteins PTEN and Ras. They introduce the notion of paradoxical network motifs and then discuss several signaling systems. Finally they discuss the implications of redox-mediated regulation of signaling pathways in brain development and disease.

This is a very well curated and written manuscript. I have one major comment, which I believe may help the authors to increase the readability and the focus of their review.

My personal opinion is that the manuscript would benefit by reducing some sections in length. Besides the discussion of a number of complicated signaling pathways that do require a lot of introductory comments (Sections 3.3-3.7), other sections include new -and somewhat lengthy - introductory parts (section 5 mitochondria, section 6 with the role of synthetic miRNAs, section 7 with the discussion of Ras/PTEN tumor suppressors with detailed description of mutation percentages,. R loops in section 7 etc). Such parts, although entirely accurate scientifically, make the text at some points extremely dense. This results in loss of clarity and focus on an otherwise extremely interesting and well-thought-out work.

Minor comments/questions to authors

1. lines 141-142: This sentence needs to be rephrased; Akt activation activates mTOR and thus inhibits autophagy.

2.  lines 188-189: Perhaps it is proper to distinguish between PI3K class I isoforms since PI3Kβ is not activated by Ras (https://doi.org/10.1016/j.cell.2013.04.031). So this module may refer primarily to PI3Kα isoform, which is the main RTK-activated PI3K.

3. line 202: This is an overstatement since PI3K activation (PIP3 generation and hence Akt activation) has been purported to occur also in endocytic vesicles (early endosomes, for example,  doi: 10.1038/s41556-020-00596-4)

4. Section 3.2: (a) Concerning PTEN there has been recent reviews on its regulation and catalytic interactions with PIP3 and membranes (e.g.  https://doi.org/10.1016/j.csbj.2022.10.007 or https://doi.org/10.1101/cshperspect.a036152, for example) which should be cited to include more recent advances. 

(b). The authors should bear in mind that PTEN has been purported to dephosphorylate itself via its protein phosphatase activity. So, besides PP2A-mediated activation, PTEN may be subject to a regulatory loop of auto-activation. Apparently, oxidation of PTEN is predicted to disrupt this loop as well.

5. lines 281-282: although published, this result has not been verified by others (according to my knowledge). Given the inherent difficulty in proving unequivocally that PTEN dephosphorylates directly Akt, i would suggest to authors to refrain from putting forward this notion openly. 

6. Section 5: Here the notion of mitochondria as a driver in the cellular clock comes a bit abruptly. Perhaps the authors should have introduced this notion/section earlier in the text.

7.  lines 589-590 and 592: This is true for PTEN but is Ras really a signal dampener? I would argue for the opposite. Similarly, Ras is not a tumor suppressor (line 592). Please rephrase.

Author Response

We would like to thank the reviewer for the encouraging comments and the constructive suggestions provided. Following is a point-by-point rebuttal that frames how the suggestions have been accommodated in the revised document.

In their review manuscript entitled "Redox-mediated rewiring of signalling pathways: The role of a cellular clock in brain health and disease" Filip Vujovic and colleagues elegantly present a novel and interesting angle on redox-mediated rewiring of signaling by focusing on two important signaling proteins PTEN and Ras. They introduce the notion of paradoxical network motifs and then discuss several signaling systems. Finally they discuss the implications of redox-mediated regulation of signaling pathways in brain development and disease.

This is a very well curated and written manuscript. I have one major comment, which I believe may help the authors to increase the readability and the focus of their review.

My personal opinion is that the manuscript would benefit by reducing some sections in length. Besides the discussion of a number of complicated signaling pathways that do require a lot of introductory comments (Sections 3.3-3.7), other sections include new -and somewhat lengthy - introductory parts (section 5 mitochondria, section 6 with the role of synthetic miRNAs, section 7 with the discussion of Ras/PTEN tumor suppressors with detailed description of mutation percentages,. R loops in section 7 etc). Such parts, although entirely accurate scientifically, make the text at some points extremely dense. This results in loss of clarity and focus on an otherwise extremely interesting and well-thought-out work.

A: We agree with reviewer’s comment. To accommodate this suggestion, we have made the following variations:

  1. We have removed all mutation percentages except in two sentences (the remaining percentage alert the reader to the frequency of these changes).
  2. We have revised, condensed, and streamlined the section on the role of R-loops in AD.

III. The section on the role of mitochondria has been revised to accommodate this comment and also a subsequent comment regarding abrupt introduction of the role of mitochondria in regulating the cellular clock.

  1. We have removed the section on cancers.

Minor comments/questions to authors

  1. lines 141-142: This sentence needs to be rephrased; Akt activation activates mTOR and thus inhibits autophagy.

A: In agreement with the reviewer, we have revised the sentence to: “Activation of PI3k/Akt via this dual signalling input complements the pro-anabolic activity of RasGTP/c-Myc axis by phosphorylation-mediated destabilisation of TSC-2 and the resultant mTOR-dependent inhibition of autophagy”.

  1. lines 188-189: Perhaps it is proper to distinguish between PI3K class I isoforms since PI3Kβ is not activated by Ras (https://doi.org/10.1016/j.cell.2013.04.031). So this module may refer primarily to PI3Kα isoform, which is the main RTK-activated PI3K.

A: Thank you for this suggestion. In the revised document, we have changed PI3K to class I PI3K and added the suggested citation. We have also made a similar change in an earlier reference to PI3K to avoid any ambiguity.

  1. line 202: This is an overstatement since PI3K activation (PIP3 generation and hence Akt activation) has been purported to occur also in endocytic vesicles (early endosomes, for example,  doi: 10.1038/s41556-020-00596-4)

A: Thank you for correcting this oversight. In the revised document, we have amended this section by stating that “Accordingly, it becomes curious whether microanatomical compartmentalisation of H-Ras GTPase to endocytic vesicles, that activates downstream Raf-1 signalling, is assisted by NADPH oxidase-dependent ROS production within the endosomal compartment to resolve the I1-FFLPI3K/Ras”.

  1. Section 3.2: (a) Concerning PTEN there has been recent reviews on its regulation and catalytic interactions with PIP3 and membranes (e.g.  https://doi.org/10.1016/j.csbj.2022.10.007 or https://doi.org/10.1101/cshperspect.a036152, for example) which should be cited to include more recent advances. 

A: In agreement with the reviewer, we have added the following statement to the section where PTEN is first introduced in the revised document: “While readers are referred to recent reviews on PTEN86,87, a summary of its regulation is provided here”. The references 86 and 87 are citations suggested by the reviewer.

(b). The authors should bear in mind that PTEN has been purported to dephosphorylate itself via its protein phosphatase activity. So, besides PP2A-mediated activation, PTEN may be subject to a regulatory loop of auto-activation. Apparently, oxidation of PTEN is predicted to disrupt this loop as well.

A: Thank you for bringing this important finding into our attention. In the revised document we have added the following statement and the associated reference: “Aside from dephosphorylation via protein phosphatase 2A, PTEN appears to be activated by auto-dephosphorylation92”.

  1. lines 281-282: although published, this result has not been verified by others (according to my knowledge). Given the inherent difficulty in proving unequivocally that PTEN dephosphorylates directly Akt, i would suggest to authors to refrain from putting forward this notion openly. 

A: In agreement with reviewer, we have removed the reference to direct dephosphorylation of Akt by PTEN, in the revised manuscript.

  1. Section 5: Here the notion of mitochondria as a driver in the cellular clock comes a bit abruptly. Perhaps the authors should have introduced this notion/section earlier in the text.

A: In agreement with this comment, we have revised the last three paragraphs of the section 4: “A blueprint for redox-mediated resetting of the cellular clock”. In the revised document, reader are gently introduced to this concept by posing a question: “A question that arises from these observations is if one considers the cellular clock as master-regulator of the pace of cellular event, what mechanism then regulates the cellular clock?”. This paragraph then leads to a discussion on the role of mitochondria as regulators of the proposed cellular clock.

  1. lines 589-590 and 592: This is true for PTEN but is Ras really a signal dampener? I would argue for the opposite. Similarly, Ras is not a tumor suppressor (line 592). Please rephrase.

A: We apologise for this oversight. The reference to dampening of signals was related to I1-FFLs and not the components of these motifs. In the revised document, we have now removed the associated sentence “and operate as signal dampeners”.

We thank the reviewer again for providing this opportunity to improve the document.

Sincerely,

Ramin M. Farahani PhD

University of Sydney/Westmead Institute for Medical Research

Reviewer 3 Report

In the review entiteled : Redox-mediated rewiring of signalling pathways: The role of a 2 cellular clock in brain health and disease the authors explained that  altered redox is generally considered as an outcome of cellular dynamics, they  review the evidence for an alternative interpretation that redox-sensitive proteins . They  suggest that a number of paradoxical motifs (I-FFLs) in the PI3K/Akt/Ras signaling pathways' network topology limit downstream signaling as a safety measure, with regulation taking place through redox-mediated rewiring of the I-FFLs. In order to achieve this, mitochondria are crucial in controlling the cellular redox state, which resets the theoretical cellular clock. Additionally, suggest that the presence of paradoxical network motifs provides a flaw into metazoan signaling pathways that become active with a protracted transition into an oxidized cellular state, leading to the principal manifestation of neurodegenerative illnesses of the human brain.

The manuscript is well organized and articulated, however I would suggest the authors to allude, even if in little depth, to the ubiquitination phenomena that are responsible for the functioning of all signal pathways, also mentioning how the alteration of these signals can affect diseases not only degenerative but also cardiac oncological etc. and how signaling targets can be used as possible targets in precision medicine. The reading is very interesting but cannot be limited to a detailed description of the already well-known signaling pathways.

 Minor editing of English language required

Author Response

We are grateful for the encouraging comments and constructive suggestions provided. Following is a point-by-point rebuttal that frames how the suggestions have been accommodated in the revised document. 

The manuscript is well organized and articulated, however I would suggest the authors to allude, even if in little depth, to the ubiquitination phenomena that are responsible for the functioning of all signal pathways, also mentioning how the alteration of these signals can affect diseases not only degenerative but also cardiac oncological etc. and how signaling targets can be used as possible targets in precision medicine. The reading is very interesting but cannot be limited to a detailed description of the already well-known signaling pathways.

A: Thank you for the provided comments. In agreement with the reviewer, we have added a new section to discuss the integration of a redox state and the ubiquitination phenomena in regulating the cellular clock. While we would like to expand the discussion to include other oncological and cardiac conditions, the manuscript will become lengthy and difficult to follow. This is reflected in comments of the reviewer 1 who suggested to declutter the document by condensing sections 4, 5, and 6 of the manuscript. In the revised document, we have expanded the section on “Reprogramming of the cellular clock: a systems biology perspective” to discuss how the concept of a cellular clock can be utilised in precision medicine. We have also changed the heading of this section to “Reprogramming of the cellular clock: application in precision medicine” in the revised document.  

We thank the reviewer again for providing this opportunity to improve the document.

Sincerely,

Ramin M. Farahani PhD

University of Sydney/Westmead Institute for Medical Research

Round 2

Reviewer 1 Report

The authors have responded adequately to my comments.

Author Response

We thank the reviewer once again for taking time to read the revised manuscript.

Best regards,

Ramin Farahani

Reviewer 2 Report

The authors have addressed all my previous comments in their rebuttal letter.

In this report I raise their attention to some typos and mistakes in the revised text.  The track changes option is in display in the pdf file and this makes it difficult to accurately assess the changes.  Apparently some changes have been mislabelled as deleted and/or have been duplicated by mistake (see for example lines 1349-1350, 1367-1382, 1433-1434, 1440-1441. I would suggest a careful proofreading of the changes before publication.

Author Response

We would like to thank the reviewer for highlighting the errors int the revised document. We have now thoroughly revised the document to remove any remaining errors. The changes are highlighted in red. Please note that some errors in the previous submission were apparently generated upon automatic conversion of the word document to the PDF format. Hence, we have only uploaded the word format. Thank you. 

Best regards,

Ramin Farahani
